# Present-day central African forest is a legacy of the 19th century human history

Julie Morin-Rivat[1,2,3*†], Adeline Fayolle[1,2*†], Charly Favier[4], Laurent Bremond[4], Sylvie Gourlet-Fleury[5], Nicolas Bayol[6], Philippe Lejeune[1,2], Hans Beeckman[3], Jean-Louis Doucet[1,2]

[1]TERRA Research Centre, Central African Forests, University of Liège – Gembloux Agro-Bio Tech, Gembloux, Belgium; [2]BIOSE, Management of Forest Resources, University of Liège – Gembloux Agro-Bio Tech, Gembloux, Belgium; [3]Wood Biology Service, Royal Museum for Central Africa, Tervuren, Belgium; [4]ISEM, Institut des Sciences de l'Évolution, UMR 5554-CNRS, Université Montpellier II, Montpellier, France; [5]Unité de Recherche Biens et Services des Écosystèmes Forestiers tropicaux, Département Environnements et Sociétés du CIRAD, Montpellier, France; [6]FRM, Montpellier, France

**Abstract** The populations of light-demanding trees that dominate the canopy of central African forests are now aging. Here, we show that the lack of regeneration of these populations began ca. 165 ya (around 1850) after major anthropogenic disturbances ceased. Since 1885, less itinerancy and disturbance in the forest has occurred because the colonial administrations concentrated people and villages along the primary communication axes. Local populations formerly gardened the forest by creating scattered openings, which were sufficiently large for the establishment of light-demanding trees. Currently, common logging operations do not create suitable openings for the regeneration of these species, whereas deforestation degrades landscapes. Using an interdisciplinary approach, which included paleoecological, archaeological, historical, and dendrological data, we highlight the long-term history of human activities across central African forests and assess the contribution of these activities to present-day forest structure and composition. The conclusions of this sobering analysis present challenges to current silvicultural practices and to those of the future.

*For correspondence: morin. rivat@gmail.com (JM-R); adeline. fayolle@ulg.ac.be (AF)

†These authors contributed equally to this work

Competing interests: The authors declare that no competing interests exist.

## Introduction

Central African forests underwent an unequal history of disturbances during the Holocene (after 10,000 yrs BP) compared with Neotropical forests, which remained relatively stable since the Late Glacial Maximum (LGM, ca. 13,000–10,000 yrs BP) (*Anhuf et al., 2006*). Over the last three millennia, significant changes in the vegetation structure and floristic composition were caused by climate fluctuations (*Maley et al., 2012*; *Neumann et al., 2012*; *Lézine et al., 2013*). Specifically, a dry event around 2500 ya caused forest fragmentationan event with a more pronounced seasonality occurred around 2500 ya and caused forest fragmentation, and this fragmented forest included patches of savanna (*Maley, 2002*). This dry episode stopped around 2500 BP, as evidenced from the Mopo Bai site in the Republic of the Congo, where Poaceae pollen severely dropped from 36% to 13% between 2580 and 2400 BP, which is evidence for a retreat of the savannas to the benefit of the forests (*Bostoen et al., 2015*). After 2000 yrs BP, a relatively wet climate in central Africa favored forest recolonization by light-demanding tree species, with few effects imputable to humans (*Maley et al., 2012*; *Lézine et al., 2013*; *Brnčić et al., 2009*; *Bostoen et al., 2015*). The subsequent climatic variations were less important with little effect on the vegetation (*Oslisly et al., 2013a*); however, human

**eLife digest** The world's forests contain trillions of trees. Some of those trees require more light than others to mature, and certain species can only grow to reach the forest canopy if they have access to sunlight throughout their whole life.

Central Africa is home to the second largest tropical rainforest in the world. Previous studies showed that few young trees of light-demanding species were growing to replace the old trees in this forest. As a result this population is aging and at risk of disappearing, which is a major concern. Many light-demanding tree species in the Central African forest are cut down for their valuable timber. However, if young trees do not grow to replace the mature ones that are logged, even logging operations that follow national and international environmental rules cannot guarantee the sustainability of these trees.

As such, Morin-Rivat et al. set out to understand what changed in the Central African forest in the past to stop the regeneration of the light-demanding trees. The analyses focused on four species classified as light-demanding trees in part of Central Africa called the northern Congo Basin. Most of the trees in these species were about 165 years old. This was the case even though the different species grow at different rates, and it means that they all grew from young trees that settled in the middle of the 19th century.

So what was it that changed after this period to stop this population of light-demanding trees in the Central African forest from regenerating? By combining information from a number of datasets and historical records, Morin-Rivat et al. arrived at the following conclusion. Before the mid-19th century, many people lived in the forest and their activities created clearings that turned the forest into a relatively patchy landscape. However from about 1850 onwards, when Europeans started to colonize the region, people and villages were moved out of the forests and closer to rivers and roads for administrative and commercial purposes. Moreover, many people were killed in conflicts or died because of newly introduced diseases, which also led to fewer people in the forest. As a result, the forest became less disturbed. With fewer clearings, fewer light-demanding trees would have had enough access to sunlight to grow to maturity.

The findings of Morin-Rivat et al. show that disturbance is needed to maintain certain forest habitats and tree species, including light-demanding species of tree. As common logging operations do not create openings large enough to guarantee that such species will be able to establish themselves naturally, complementary treatments are needed. These might include selectively logging mature trees around young members of light-demanding species, or planting threatened species.

activities are assumed to have increased in importance, particularly during the most recent centuries (*Oslisly et al., 2013a*, 2013b; *Willis et al., 2004*; *Brnčić et al., 2007*; *Greve et al., 2011*). The abundance of direct (artifacts) and indirect evidence (charred oil palm endocarps) in soils confirms the non-pristine nature of central African forests (*Morin-Rivat et al., 2014*).

Human activities in the Holocene, and particularly shifting cultivation, have been invoked to partially explain the low diversity of central African forests (*Parmentier et al., 2007*) and the abundance of light-demanding species in the canopy (*White and Oates, 1999*; *van Gemerden et al., 2003*; *Engone Obiang et al., 2014*; *Vleminckx et al., 2014*; *Biwolé et al., 2015*). The light-demanding species form, in some places, almost pure 0.5 to 1 ha stands that mirror the size of traditionally cultivated fields (*van Gemerden et al., 2003*). An example is the Sangha River Interval (SRI) in which the vegetation currently forms a 'corridor' of old-growth semi-deciduous *Celtis* forests (*Fayolle et al., 2014a*; *Gond et al., 2013*), with local variations caused by the geological substrate or the forest degradation along roads and close to cities (*Fayolle et al., 2012*) (*Figure 1*). The SRI is a 400-km-wide region, with low endemism between the Lower Guinean and the Congolian subcenters of endemism (*White, 1983*). This area, which is between southeastern Cameroon, southern Central African Republic and northern Congo, may have been a savanna corridor 2500 ya (*Maley, 2002*). Until the recent studies of *Harris (2002)*, and *Gillet and Doucet (2012)*, the vegetation in the SRI was under sampled, and whether the origin of this corridor is environmental (*Fayolle et al., 2012*) or

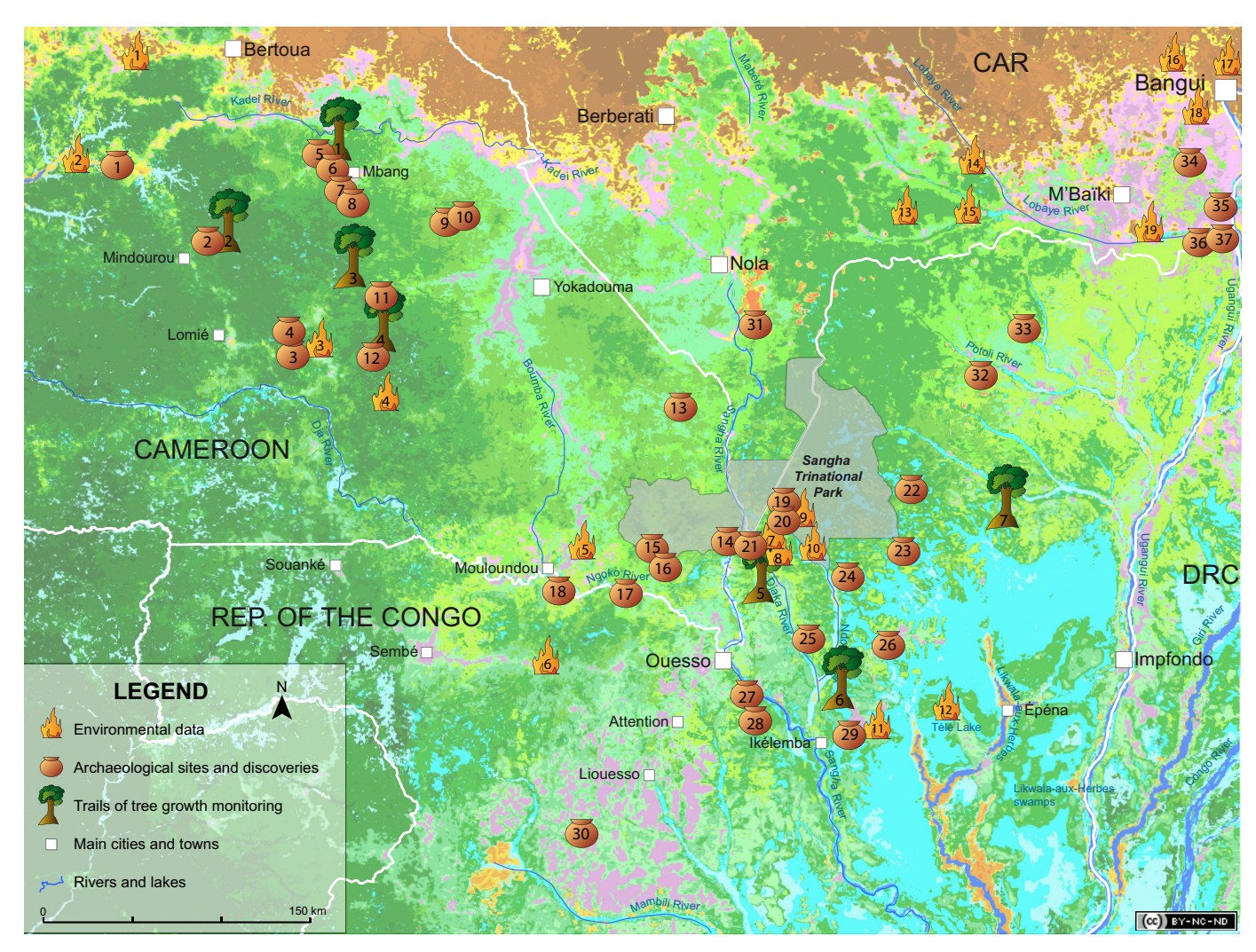

**Figure 1.** Paleoenvironmental changes and human activities in the Sangha River Interval. The 34 sites with paleoenvironmental data (fires) and the 38 dated archaeological sites and discoveries (pots) are indicated on a vegetation map modified from *Gond et al. (2013)* (http://www.coforchange.eu/products/maps). The seven sites used to monitor tree growth (trees) are also indicated (see *Supplementary files 4*, *5* and *6* for site names). Brown (three shades): savanna of the Sudano-Guinean domain; orange (three shades): savanna included in dense forest; yellow: savanna-forest edge; purple (two shades): very open forest; blue-green: open semi-deciduous forest; medium green (three shades): dense semi-deciduous forest; dark green (five shades): dense evergreen forest; light green (two shades): open evergreen forest; light blue (two shades): swamp forest and swamp. Map: QGIS 2.14 (http://www.qgis.org), CAD: Illustrator CS4 (https://www.adobe.com).

historical (*Morin-Rivat et al., 2014*) remains to be explored. In this study, we assessed the potential impact of historical human activities on central African forests. Specifically, we analyzed the population/age structure of four primary light-demanding timber species across the SRI and examined the synchronism with the paleoenvironmental, archaeological, and historical data in this region (*Figure 1*).

# Results

## Forest composition

The 1,765,483 inventoried trees were studied at the genus level, and included 176 genera (*Supplementary file 1*). The five most represented genera were *Celtis* (Ulmaceae), *Polyalthia* (Annonaceae), *Strombosia* (Olacaceae), *Petersianthus* (Lecythidaceae), and *Manilkara* (Sapotaceae).

Most of the genera included shade-bearers (n = 71 genera), which were followed by the pioneers (n = 47), and the non-pioneer light-demanding species (NPLD, n = 37). We had no information for 21 genera. Regarding leaf phenology, 108 genera were evergreen, versus 50 deciduous. No information was available for 16 genera.

Wood density ranged from 0.22 g.cm$^{-3}$ for *Ricinodendron* (Euphorbiaceae) to 0.88 g.cm$^{-3}$ for *Bobgunnia* (Fabaceae). Mean density was 0.58 g.cm$^{-3}$. Mean diameters ranged from 31.62 cm to 93.46 cm in dbh for *Meiocarpidium* (Annonaceae) and *Autranella* (Sapotaceae), respectively, with a mean for all genera of 47.45 cm in dbh. Mean basal area ranged from 0.12 m$^2$ to 0.92 m$^2$ for *Lasiodiscus* (Rhamnaceae) and *Ceiba* (Malvaceae), respectively, with a mean for all genera of 0.30 m$^2$.

## Forest structure

Among the inventoried trees, we identified two groups of genera: (i) those that showed a reverse-J shape distribution (*Figure 2*, and *Supplementary file 1*) with many small and young trees (most of the genera, n = 134, 76%), and (ii) those for which distributions deviated from this pattern (n = 42, 24%), including flat (e.g., *Baillonella*) and unimodal distributions of diameter. Among these, we identified four primary canopy genera (i.e., *Erythrophleum* and *Pericopsis* (Fabaceae), *Terminalia* (Combretaceae), and *Triplochiton* (Malvaceae)) with unimodal diameter distributions (*Figure 2—figure supplement 1*).

These genera are monospecific in the SRI (*Pericopsis elata*, *Terminalia superba*, *Erythrophleum suaveolens*, and *Triplochiton scleroxylon*), and share similar functional traits (i.e., deciduous, emergent, pioneer light-demanding trees). Combined, these four species represented 4.3% of the inventoried trees, reaching a maximum of 8.62% in one site in Cameroon.

## Diameter distribution of the four studied species

The dbh ranged from 10.6 cm (*T. superba*) to 151.6 cm (*E. suaveolens*) (*Supplementary file 2*). The mode of the diameter distribution differed between the four studied light-demanding species, with 65.3 cm for *P. elata*, 69.8 cm for *T. superba*, 72 cm for *E. suaveolens*, and 90.3 cm for *T. scleroxylon*. Weibull distributions indicated modes comprised between 65.3 cm in dbh for *P. elata*, and 90.3 cm in dbh for *T. scleroxylon*. The modes for *T. superba* and *E. suaveolens* were 69.5 cm and 72 cm in dbh, respectively.

## Tree-ring data

Four studies provided growth and age data, which were based on tree-ring analysis (*Supplementary file 3*). We found data for 83 discs (*P. elata* = 24; *T. superba* = 41; *T. scleroxylon* = 18) from four locations in the Democratic Republic of Congo, the Ivory Coast, and Cameroon. Data for *E. suaveolens* were not available. Mean ring width ranged from 0.298 ± 0.54 cm for *P. elata* to 0.719 ± 0.267 for *T. superba*. It was 0.620 ± 0.28 cm for *T. scleroxylon*.

## Mean annual increment in diameter (MAI$_d$)

In the study sites, the MAI$_d$ of the monitored trees ranged from 0.44 ± 0.033 cm/y for *E. suaveolens* (367 stems) to 0.58 ± 0.061 cm/y for the fast-growing *T. scleroxylon* (265 stems). It was 0.45 ± 0.026 cm/y and 0.53 ± 0.112 cm/y for *P. elata* (199 stems) and *T. superba* (152 stems), respectively (*Supplementary file 2*).

## Performance of the growth models

Results of tree modeling (*Figure 2—figure supplement 2*, and *Supplementary file 4*) indicated that the Canham model was the best model to explain tree growth in *E. suaveolens* (BIC = 196.6), *T. superba* (BIC = 256.1), and *T. scleroxylon* (BIC = 372.1), whereas only the Mean model best

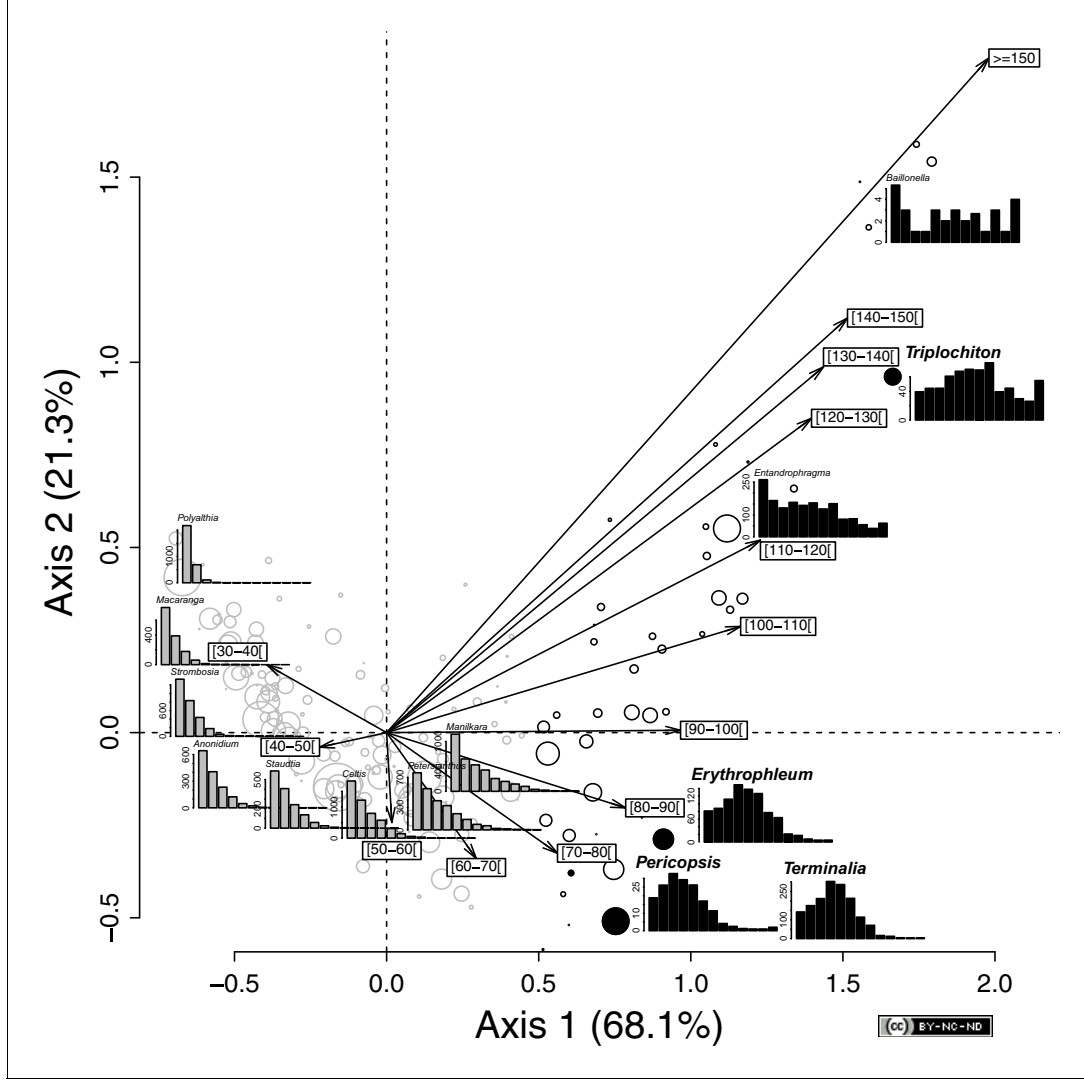

**Figure 2.** Variation in tree diameter distribution among the 176 genera across the SRI. Projection of the genera and the 10-cm-wide diameter classes in the ordination space defined by the first two axes of a correspondence analysis of the abundance matrix, as defined by 176 genera and 13 diameter classes. The size of the circles is proportional to the square root of the genus abundance. The color of the symbol corresponds to the two groups identified with a clustering analysis (based on Euclidean distances and an average agglomeration method) on the species score on the first factorial axis. Genera that showed a reverse-J diameter distribution (n = 134) are indicated in gray and those genera that showed a deviation from the reverse-J distribution (n = 42) in black (e.g., *Baillonella*). Black filled circles indicate the four genera that are monospecific in the SRI and used for the age estimations. Diameter distribution of the 10 most abundant genera is shown in addition to that of the four selected genera: *Celtis* (gray), *Polyalthia* (gray), *Strombosia* (gray), *Petersianthus* (gray), *Manilkara* (gray), *Entandrophragma* (black), *Terminalia* (black), *Anonidium* (gray), *Staudtia* (gray), and Macaranga (gray). Statistics: R (https://www.r-project.org/), CAD: Illustrator CS4 (https://www.adobe.com).

The following figure supplements are available for figure 2:

**Figure supplement 1.** Distribution of diameters of the four study species in the 22 study sites (black).

**Figure supplement 2.** Growth models (a, c, e and g) and growth trajectories (b, d, f and h) for the four study species based on tree-ring data.

explained tree growth in *P. elata* (BIC = −99.1). The performance of the models remained, however, very low.

## Growth/age relationship

According to the age data from published tree-ring studies (*Supplementary file 3*), we found that estimations based on mean growth were likely to be more reliable than those based on growth models (*Figure 2—figure supplement 2*). In particular, the performances of the Canham and Lognormal models were low, as well as, to a lesser extent, that of the unimodal distributions (sigmoidal growth trajectory). Based on mean growth estimates, the age of the canopy trees was only a few centuries, with a mode dated to between 142 and 164 ya, which corresponds to the years AD 1836 and 1858 (mean AD 1850) (*Figure 2*).

## Chronology of paleoenvironmental changes

Climate of the last 1000 years was documented by sea surface temperatures (SSTs) and the atmospheric dust signal from the marine core ODP 659, taken off the West African coast, and sediments from Mopo Bai and Goualogou Lake in the Republic of the Congo. Climate oscillated between wet and dry periods (*Figure 3*, and *Supplementary file 5*). Typically, climate was dry until ca. AD 1200, between AD 1250 and 1450, and since AD 1850, with intermediate wet periods, in particular a long one between ca. AD 1450 and 1850.

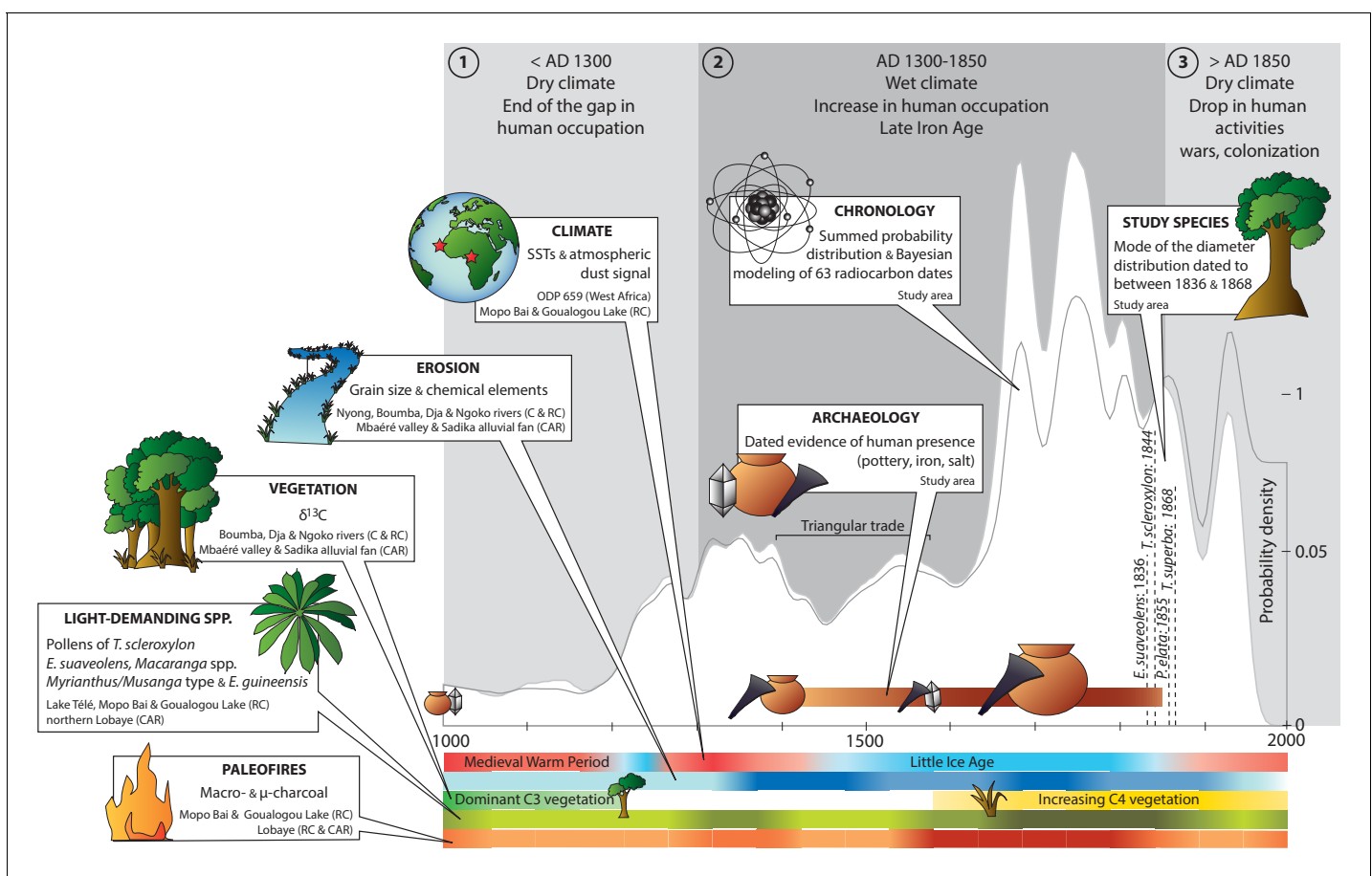

**Figure 3.** Chronology of paleoenvironmental changes and human activities in the Sangha River Interval. We compiled data on climate, erosion, vegetation types, light-demanding species and paleofires for the last 1000 years from 34 paleoenvironmental sites and data from 38 dated archaeological sites and discoveries with 63 related radiocarbon dates. The summed probability distribution of the radiocarbon ages showed fluctuations in the signal of human activities through time. Three primary time periods were identified: (a) before AD 1300; (b) from AD 1300 to 1850; and (c) after AD 1850. Color scales (four levels) were assigned depending on the proxy influx on the curve: light = present but rare; light-medium = present; medium = frequent; and dark = very frequent. Abbreviations: AD = Anno Domini (= calendar dates); C = Cameroon; RC = Republic of the Congo; CAR = Central African Republic; SSTs = Sea Surface Temperatures; C3/C4 plants = woody species (below −20‰)/herbs (above −20‰); *E. guineensis* = the oil palm *Elaeis guineensis* (*Supplementary files 5* and *6*). CAD: Illustrator CS4 (https://www.adobe.com).

The erosion curve included data related to grain size and chemical elements from the banks of the Nyong, Boumba, Dja, Ngoko rivers in Cameroon and in the Republic of the Congo, and the Mbaéré valley and the Sadika alluvial fan in the Central African Republic. This signal did not overlap climate data, as erosion was high between ca. AD 1350 and 1950, with a slight drop dated to between AD 1500 and 1650.

The history of vegetation change derived from $\delta^{13}C$ values obtained at the same sites as those documenting erosion. Results indicated two main time periods: a first one until ca. AD 1200 dominated by forest vegetation (C3 dominant, values >25‰), and a second one from ca. AD 1600 until today dominated by grass cover (C4 dominant, values <25‰).

Pollen data of light-demanding species (i.e. *T. scleroxylon*, *E. suaveolens*, *Macaranga* spp., *Myrianthus/Musanga* type and *E. guineensis*) were obtained at Lake Télé, Mopo Bai and Goualogou Lake in the Republic of the Congo, and at sites in the northern Lobaye in the Central African Republic. They were more present between ca. AD 1300 and 1400, then between ca. AD 1600 and 1850.

Paleofires were documented by macro- and microcharcoal data from Mopo Bai and Goualogou Lake in the Republic of the Congo, and the Lobaye area (in the Rep. of the Congo and the Central African Republic). Indicators of paleofires slightly increased between ca. AD 1300 and 1400. They were more substantial, however, between ca. AD 1550 and 1850.

## Chronology of human activities

Evidence of human activities was identified during two main periods: the first one around AD 1000, and the second between ca. AD 1400 and 1850 (*Figure 3*, and *Supplementary file 6*). The main discoveries comprised potsherds associated with settlements, iron slags and *tuyères* related to iron smelting, or were located in places where salt was exploited (e.g. Ngoko River). Most of the artifacts were found between ca. AD 1600 and 1800. Focusing on the pottery only, dates were distributed into three periods: (i) between ca. AD 800 and 1100, (ii) between AD 1300 and 1600, and (iii) between AD 1700 and 1800. Smelting activities were documented at a few sites only, especially in the southern Central African Republic (i.e. Bagbaya, Ngara, and Lingbangbo), which were in use during short time periods: between ca. AD 1000–1100, AD 1300–1400, AD 1500–1700, and AD 1700–1900.

## Radiocarbon chronology

The results of the Bayesian analysis of the radiocarbon dates indicated a weak radiocarbon signal until ca. AD 1200, which increased from ca. AD 1200 (*Figure 3*, and *Supplementary file 7*). Main peaks were centered on ca. AD 1350, 1550, and 1750. The signal strongly decreased after ca. AD 1800, with a last small peak around AD 1950 related to the nuclear activities of the mi-twentieth century.

## Historical events

Key events emerged within the historical chronology (*Supplementary file 8*). Firstly, the Triangular Trade, and particularly the period between AD 1400 and 1600, profoundly destabilized the area. During the following centuries, the slave-raiding, leaded by the Fulbe people, pushed other populations to flee southward in the forest.

The second key event is the beginning of the colonization of Africa, which put a stop to the Fulbe's activities. The exploration of the SRI that began after AD 1875, and the permanent presence of the European colonists since then, deeply disturbed the spatial distribution of the local populations, as well as their activities (e.g. enrolment in the concession companies, education, diseases, etc.). During this period, the conflicts that opposed France and Germany (i.e. the 1870 War, and the First and Second World Wars) were also transferred to the African territories.

Finally, the region experienced a massive rural exodus since the 1930s, which was amplified since the independences (Cameroon, Republic of the Congo, and Central African Republic the same year: 1960).

## Discussion

### Generalized decline of light-demanding tree populations

The reverse-J-shape distribution of diameters, characteristic of most genera, is typical of 'active' tree populations with many small and young trees (*Figure 2—figure supplement 1*). By contrast, the unimodal distribution of diameters for could represent a generalized limited number of young trees (i.e., a lack of regeneration) and indicate the widespread decline of the tree populations. This type of distribution was characteristic of four primary canopy genera (i.e., *Erythrophleum* and *Pericopsis* (Fabaceae), *Terminalia* (Combretaceae), and *Triplochiton* (Malvaceae)), which we studied further. Historical factors were previously invoked to explain such distributions for *E. suaveolens* and *T. superba* in eastern Cameroon (*Durrieu de Madron and Forni, 1997*). Similarly, a unimodal distribution of diameters was reported for the light-demanding timber species *Aucoumea klaineana* in Gabon, which could not be explained only by demography.

Ontogenic variations in growth are well described for tropical tree species, and unimodal growth trajectories are widely reported (*Hérault et al., 2011*). The low performance of the models to estimate tree age is explained by the slow growth of young trees (dbh ≤10 cm) and the great uncertainty regarding the time a tree remains in the small diameter classes (*Figure 2—figure supplement 2*, and *Supplementary file 4*). Indeed, a linear relationship between tree diameter and age is acceptable for tropical tree species of a larger size (*Worbes et al., 2003*). Most suppressed individuals were destined to die, and therefore, only the trees with vigorous growth are able to reach the canopy and could be thus included in this type of analysis. The tree-ring approach, including information for the growth of small trees, remains therefore essential for age estimation (*Worbes et al., 2003*), but studies are only sporadic for central African forests.

Based on mean growth estimates, canopy trees in the SRI were aged to only a few centuries, with a mode dated to AD 1850 in average. This age range is consistent with the estimated ages of canopy trees in Nigeria (*van Gemerden et al., 2003*) and in Cameroon (*Worbes et al., 2003*). Moreover, the population decline of *A. klaineana* in Gabon is attributed to a shift in the disturbance regime two to three centuries ago (*Engone Obiang et al., 2014*). The argument for a regional trend is supported by these age estimates and the general pattern we reported across the SRI. We assumed that the unimodal population/age structure of the light-demanding tree species was linked to the recent human history. Specifically, we postulated that the decrease in anthropogenic disturbances and the generalized land abandonment from ca. 165 ya were less favorable to the regeneration of light-demanding tree species (*van Gemerden et al., 2003*; *Brnčić et al., 2007*; *Greve et al., 2011*; *Biwolé et al., 2015*). Additionally, the present-day natural gap size has been shown to be insufficient for the regeneration of most of these species (*van Gemerden et al., 2003*).

### The regional history of human activities

All proxies converged toward the identical regional history that is divided into three primary periods: (i) a dry period between AD 950 and 1300 with almost no human activity recorded; (ii) a wet period between AD 1300 and 1850 with large-scale human activities and a high disturbance regime that led to a forest-savanna mosaic; and (iii) a forest aging period from AD 1850 to the present. The aging period corresponded to a shift in the disturbance regime that was most likely caused by a depopulation of the forest with the beginning of the European colonization (*Robineau, 1967*; *Copet-Rougier, 1998*; *Coquery-Vidrovitch, 1998*).

The first time period before AD 1300 corresponds to a dry climate, consistent with the higher latitude Medieval Warm Period (*DeMenocal et al., 2000*), with only scarce pollen of pioneer and light-demanding species (*Brnčić et al., 2007, 2009*). The vegetation was composed of forest tree species according to $\delta^{13}C$ values between −30.6 and −25.8‰ (*Sangen, 2012*). In southeastern Cameroon, alluvial records indicate a growing human impact on forests between AD 1000 and 1200, particularly because of shifting cultivation and the associated increase in erosional processes (*Sangen, 2012*; *Runge et al., 2014*). Charcoal (related to natural fires and anthropogenic burning) in lake sediments and soils were recorded only at the end of this period, which corresponds to the end of the hiatus phase in human activities (massive depopulation) previously documented for central Africa (*Oslisly et al., 2013a, 2013b*; *Wotzka, 2006*) and specifically for the SRI (*Morin-Rivat et al., 2014*). Surveys have been carried out, but the poor surface visibility can underestimate true human

presence (see *Morin-Rivat et al., 2016*, for an example of methodology). Sometimes no archaeological research have been carried out in certain regions covered by dense forest (B. Clist, pers. comment). Notably, at this time, human populations were only indicated at a few sites that were dedicated to iron metallurgy (southern CAR near Bangui and Nola and the site of Ngombé in the Rep. of the Congo, approximately AD 1300) and to salt exploitation (Ngoko River, approximately AD 1000).

From AD 1300 to 1850, pollen sequences indicated a relatively wet climatic period. Nevertheless, burning increased, and this burning is attributed to human activities because the moisture content of the vegetation was too high for fires to often occur naturally (*Brnčić et al., 2007*, *2009*; *Vennetier, 1963*). The Mbaéré valley and the Sadika alluvial fan (Gadzi-Carnot sandstones in CAR) recorded intensive erosion and relatively high $\delta^{13}C$ values after AD 1200, indicating forest regression and the formation of a forest-savanna mosaic (*Sangen, 2012*; *Runge et al., 2014*). In southeastern Cameroon, the anthropogenic erosion culminated at approximately AD 1200–1400 (*Sangen, 2012*; *Runge et al., 2014*). The decrease in the run-off with an increased rate of sedimentation between AD 1400 and 1600 corresponds to the climatic period of the Little Ice Age (*Brnčić et al., 2007*, *2009*), in combination with an increase in the frequency of El-Niño events between AD 1200 and 1500 (*Sangen, 2012*). Since then, despite a more humid period following the Little Ice Age, maximal incidence of human activities have been recorded in the SRI, which opened the forest cover and favored the pioneers. Nonetheless, we must remain cautious regarding the interpretation of the archaeological data, as there is a huge gap of knowledge in the SRI, especially in the area between Souanké and Berberati (*Figure 1*). In the state of the art, it is not possible to interpret the spatial distribution of human settlements and activities. In particular, iron-smelting sites are only few, they are concentrated in the southern Central African Republic, and were in use during short periods. The volume of charcoal used and, by extension, the associated deforestation, should have been important for feeding the furnaces, as shown by *Pinçon (1990)*. However, the debate is still alive about estimating the volume of wood needed for metallurgy (*Lupo et al., 2015*), compared to the volume of trees logged for shifting agriculture (*Goucher, 1981*).

The period of ca. AD 1850 to the present marked a decrease in the disturbance regime (*Figure 3*). The pollen of naturally grown oil palms and pioneer trees became rare or absent. In southeastern Cameroon and in the CAR, pollen, phytoliths, soil charcoal and $\delta^{13}C$ values indicate little disturbance during the past 100–150 years, with the recolonization of the savannas by forest trees (*Runge et al., 2014*; *Lupo et al., 2015*). The anthropogenic burning persisted, as indicated by charcoal particles found in sites located along rivers (*Brnčić et al., 2007*, *2009*; *Tovar et al., 2014*), which might document either the colluvium of charcoals downslope or the concentration of human activities on the riverbanks. During this period, less evidence of human activities is reported (*Oslisly, 2013b*; *Morin-Rivat, 2014*). In the 1960s, young secondary forests (i.e., with *Musanga cecropioides*) constituted only 1% of the forest types and were located along the main roads (*Vennetier, 1963*). Despite the drying of the 20th century, confirmed by low flow regimes in the primary rivers, the Sangha, Ubangui, Lobaye, and Likwala-aux-Herbes (*Runge and Nguimalet, 2005*; *Aleman et al., 2013*), and the increase in anthropogenic activities in recent years (e.g., mining, industrial logging from the 1970s, burning, and cultivation) that induced very localized, degraded landscapes (*Laporte et al., 2007*; *Sangen, 2012*; *Gond et al., 2013*), forests apparently extended naturally in central Africa (*Sangen, 2012*).

## Recent and generalized land abandonment

Although precise historical information is not available before the mid-19th century for central Africa (*Burnham, 1996*; *Robineau, 1967*) (see *Supplementary file 8* for a detailed chronology), we observed a drop in the radiocarbon signal between AD 1400 and 1650 (*Figure 3*) that we assigned to the inland impacts of the Triangular trade in the late 15th century (*Gendreau, 2010*). Indeed, between AD 1550 and 1850, the Fulbe populations coming from northern Cameroon (*Burnham, 1996*) organized the slave-raiding for Europeans and induced the flight of populations southward into the forest (*Vennetier, 1963*), explaining the increase in human presence and activities (i.e. agriculture and smelting) in the region. The successive displacements of groups until the 18th century explain the numerous interethnic wars in the Upper-Sangha, for land control and cultural supremacy (*Copet-Rougier, 1998*).

Based on the large dataset that we gathered, human activities clearly decreased after ca. AD 1850, which corresponds to the beginning of the regeneration shortage of light-demanding tree populations. In the last decades of the 19th century, Savorgnan de Brazza reported that the SRI was densely populated (*Copet-Rougier, 1998*), which seems now unlikely given the low density of human populations (less than one inhabitant per km$^2$). We hypothesize that the European colonization deeply disturbed the spatial organization of the local populations in central Africa, as demonstrated in Gabon (*Pourtier, 1989*; *Engone Obiang et al., 2014*). Colonization stopped the migrations and the interethnic warfare and forced entire groups to settle along rivers and roads for administrative and commercial purposes (*Vennetier, 1963*; *Robineau, 1967*). However, the process of village redistribution during the colonial times strongly varied from one place to another, according to the settlement of the colonial posts, and the borders between the French and German possessions (*Pourtier, 1989*: e.g. of the Fang villages in Gabon). Additional factors can also be invoked to explain the emptying of the forests, including the involvement of local populations in the Franco-German conflicts during their respective colonial expansions and the two World Wars, the forced or voluntarily labor in concession companies, the deadly repression of riots, and the increased mortality because of diseases (e.g., trypanosomiasis along the Ubangui and the Sangha rivers) (*Robineau, 1967*; *Runge and Nguimalet, 2005*; *Runge, 2008*; *Runge et al., 2014*). Furthermore, because of the land abandonment caused by the new relationships established between the local peoples and the colonists (*Giles-Vernick, 2000*), the Mpiemu tales of the late 19th century relate to the regrowth of the forest.

From the 1930s and after the independence (1960), the abandonment of the forests was amplified because the access to education contributed to an increase in the rural exodus to the main towns and capitals in a search for valued wage labor in administration or trade (*Vennetier, 1963*; *Robineau, 1967*). From this period, deep demographic disparities emerged between towns and rural areas: most working-age people went to cities (e.g., Ouesso, Impfondo and Brazzaville in Congo, Yokadouma and Bertoua in Cameroon, and Berberati and Bangui in CAR), while children and the elderly people were left in villages. Thus, less labor force was available for forest clearing and cultivation (*Vennetier, 1963*).

## Conclusion

For the first time in the Sangha River Interval, a convergent body of evidence shows the effect of past changes in the disturbance regime on forest structure and composition. Consistent with previous observations in Nigeria (*White and Oates, 1999*; *van Gemerden et al., 2003*), in Gabon (*Engone Obiang et al., 2014*), and in southwestern Cameroon (*Biwolé et al., 2015*), the population decline of light-demanding tree species that now dominate the canopy is explained by the decrease in anthropogenic disturbances. Caution is nevertheless required regarding the interpretation of the radiocarbon signal. Large-scale historical events, such as the interethnic wars and the European colonization of Africa, contributed to reduce human pressure on the forest. Former agricultural activities such as shifting cultivation, which were scattered in the forest areas between AD 1300 and 1850, likely had an indirect positive influence on the regeneration of these species. Past local populations of 'foragers-horticulturists' (*Kay and Kaplan, 2015*) gardened the forest by preserving useful light wooded trees (e.g., *T. scleroxylon*) or dense wooded trees (e.g., *P. elata* and *E. suaveolens*) in the fields during forest clearing and therefore created favorable conditions for their recruitment (*Carrière et al., 2002*). Since ca. AD 1850, the reduced disturbance regime has apparently hindered the regeneration of most species of light-demanding trees (*Carrière et al., 2002*; *van Gemerden et al., 2003*; *Willis et al., 2004*; *Brnčić et al., 2007*). The current lack of regeneration and the general aging of the populations threaten both their viability and the sustainability of logging (*Hall et al., 2003*; *van Gemerden et al., 2003*). Thus, based on these results, a renewed interest in silvicultural practices (*Doucet et al., 2004*) that create larger openings in the canopy should be inspired. Complementary liberation, thinning treatments, and population enforcement, may also contribute to maintain these timber species (*Fayolle et al., 2014a*).

## Materials and methods

### Study area

The Sangha River Interval (SRI) is a 400-km-wide area in southeastern Cameroon, southern Central African Republic (CAR), and northern Republic of Congo. The extremes that encompass the area are 0°−5° N and 13°−19° E (*Gond et al., 2013*). The climate is humid tropical to equatorial from north to south and from east to west with alternating wet (May, September-October) and dry seasons (December-February, July; *Gillet and Doucet, 2012*). Mean annual rainfall ranges between 1616 and 1760 mm (Lomié in Cameroon and Impfondo in the Republic of the Congo; www.climatedata.eu). Monthly average temperatures fluctuate around 25°C. The vegetation of the area corresponds to moist forests of the Guineo-Congolian domain (*White, 1983*; *Gond et al., 2013*; *Fayolle et al., 2014b*).

### Forest inventory data

We used published analyzed forest inventory data (*Fayolle et al., 2014a*) from 22 sites (i.e., forest concessions) scattered over southeastern Cameroon (n = 6), southeastern Central African Republic (n = 6), and northern Republic of Congo (n = 10) (*Supplementary file 1*). The forest inventories were conducted between 2000 and 2007 with a systematic sampling ≥1% of the concession area. We used a dataset with 1,765,483 inventoried trees with a dbh ≥30 cm in 22 sites (i.e., forest concessions before exploitation) that covered six million ha in the SRI (*Fayolle et al., 2014a*) (*Supplementary file 1*). We examined the diameter distribution at the genus level for the entire SRI. All trees ≥30 cm in diameter at breast height (dbh) were identified and measured in 0.5 ha plots consecutively distributed along parallel and equidistant transects in unlogged forest concessions (*Picard and Gourlet-Fleury, 2008*; *Réjou-Méchain et al., 2008*; *Gourlet-Fleury et al., 2011*; *Gond et al., 2013*; *Fayolle et al., 2012, 2014a*). The minimum diameter of the trees recorded was 30 cm, which effectively confined our analysis to (sub)canopy trees with reduced mortality and less variation in growth rates (*Clark and Clark, 1992*). Vernacular names were converted into genus-level scientific names, and the trees were assigned to 10-cm-wide diameter at breast height (dbh) classes, with the largest trees ≥150 cm in a single class (total of 13 classes). Diameter distributions were analyzed for a set of 176 of the inventoried genera for which we were confident of the identification (*Fayolle et al., 2014a*).

### Analysis of diameter distribution

To detect the main variation in the diameter distribution among the genera, we performed a correspondence analysis (CA) of the genus × diameter matrix followed by a clustering based on Euclidian distances and an average agglomeration method. In this study, we focused on four particular genera that are monospecific in the SRI and had a unimodal distribution (*Figure 2* and *Supplementary file 1*) and for which we had data on their annual increments of diameter (i.e., *Erythrophleum*, *Pericopsis*, *Terminalia*, and *Triplochiton*). Details on the diameter distribution of the study species at each study site are shown in *Figure 2—figure supplement 1*. *Terminalia* and *Triplochiton* are characteristic of semi-deciduous *Celtis* spp. forest in the SRI (*Fayolle et al., 2014b*), whereas *Pericopsis* is an endangered timber species according to the CITES Red List. We later refer to species only (i.e., *Erythrophleum suaveolens* and *Pericopsis elata*, *Terminalia superba*, and *Triplochiton scleroxylon*) as they are monospecific in the study area.

### Published age data

We gathered age data for the four study species in tropical Africa from published tree-ring studies (*Worbes et al., 2003*; *De Ridder et al., 2013a, 2013b, 2014*) (*Supplementary file 3*) to identify the growth models that provided reliable age estimations (*Figure 2—figure supplement 2* and *Supplementary files 3* and *4*). All trees were measured at dbh (130 cm in height). In *Figure 2—figure supplement 2*, the age/diameter relationships are shown.

### Growth data

Repeated diameter measurements of 982 monitored trees of the four study species were obtained on seven trails (n = 4 in Cameroon; n = 3 in the Republic of the Congo) used for the permanent

monitoring of tree growth (*Picard and Gourlet-Fleury, 2008*). We calculated the mean annual increment in diameter (MAI$_d$) for n = 367 *E. suaveolens*; n = 199 *P. elata*; n = 152 *T. superba*; and n = 264 *T. scleroxylon*.

## Growth models

To account for the ontogenic variation in growth generally identified for tropical tree species (*Hérault et al., 2011*), six growth models (i.e., Canham, Gompertz, Verhulst, Power, Power mult, and Lognormal) relating tree diameter (DBH) to growth (MAI$_d$) were fitted to the growth and diameter data for all study species. Linear and Mean models were additionally fitted for comparison (*Supplementary file 4* and *Figure 2—figure supplement 2*). We used the Bayesian Information Criterion (BIC) for assessing the performance of the models.

## Age estimation

Ordinary differential equations were solved numerically to obtain the relationship between tree diameter and time (age) (*Figure 2—figure supplement 2*). We finally estimated the age of trees at the mode of the diameter distribution based on the Mean Annual Increment of diameter (MAI$_d$) and converted these ages into dates using the inventory date of AD 2000 as the reference date (*Supplementary file 2*).

## Synthesis of paleoenvironmental changes

We documented the paleoenvironmental changes in the SRI for the last 1000 years (*Supplementary file 5*) (*Laraque et al, 1998*; *DeMenocal et al., 2000*; *Runge and Fimbel, 2001*; *Harris, 2002*; *Runge and Nguimalet, 2005*; *Brnčić et al., 2007*, *2009* ;*Neumer et al., 2008*; *Runge, 2008*; *Sangen et al., 2011*; *Sangen, 2012*; *Aleman et al., 2013*; *Runge et al., 2014*; *Tovar et al., 2014*; *Lupo et al., 2015*). We acquired paleoenvironmental data from 34 sites, either *terra firme*, swamp, lake or marine sites, that provided data on the past climate (SSTs and atmospheric dust signal), vegetation (phytoliths, $\delta^{13}$C, pollen) and anthropogenic disturbances (charcoal influxes, alluvial discharges through grain size and chemical elements analyses). Site locations are shown in *Figure 1*, and the data are synthesized in *Figure 3*. The degree of frequency of a proxy was determined regarding all similar curves in the identical study (e.g., *E. guineensis* pollen curve ~ all pollen curves in *Brnčić et al., 2009*), and the cutoffs were evenly set from the minimum to the maximum values.

## Synthesis of human activities

We used 63 uncalibrated traditional and accelerator mass spectrometry (AMS) radiocarbon dates and two optically stimulated luminescence (OSL) dates from 52 archaeological sites and punctual discoveries that covered the last 1000 years (*Supplementary file 6*) (*Fay, 1997*; *Lanfranchi et al., 1998*; *Brnčić, 2003*; *Moga, 2008*; *Meyer et al., 2009*; *Oslisly et al., 2013b*; *Morin-Rivat et al., 2014*, *2016* ;*Lupo et al., 2015*). A total of 22 published dates from 21 sites in Cameroon, 15 dates from 13 sites in the Republic of the Congo, and 28 dates from 18 sites in the Central African Republic were acquired. The site locations are shown in *Figure 1*.

## Bayesian analysis of the radiocarbon dates

The analyses on dates were performed using the OxCal v.4.2 program (*Ramsey, 2013*) with the IntCal13 atmospheric calibration curve (*Reimer, 2013*). All dates were tested using an outlier analysis (*Ramsey, 2009*). To provide an estimate of the temporal trends of human activities in the SRI, we performed a summed probability distribution of the 63 available radiocarbon dates calibrated in yrs BP in combination with a Bayesian model (*Bayliss, 2009*; *Ramsey, 2009*) (*Figure 3*). Chronological Query Language (CQL) codes used are indicated in the *Supplementary file 7*.

## Synthesis of the historical data

After reviewing the historical literature, we selected 12 references that illustrate key dates and events from the beginning of the 15th century to the present, which influenced directly or indirectly human populations in the SRI (*Supplementary file 8*) (*Vennetier, 1963*; *Robineau, 1967*; *Kaspi, 1971*; *Burnham, 1996*; *Copet-Rougier, 1998*; *Coquery-Vidrovitch, 1998*; *Freed, 2010*;

*Giles-Vernick, 2000*; *Manning and Akyeampong, 2006*; *Laporte et al., 2007*; *Gendreau, 2010*; *Stock, 2013*). All cited localities are indicated in *Figure 1*.

## Acknowledgements

The CoForChange project (EraNet-Biodiversa - ANR/NERC) '*Predicting the effects of global change on forest biodiversity in the Congo Basin*', the *King Léopold III Fund for Nature Exploration and Conservation* (Belgium), and the FRFC project (F.R.S./FNRS, No. 2.4577.10) '*Dynamics of light-demanding tree species and grasses in the humid forests of Central Africa in relationship with past anthropogenic and climate disturbance*' supported this study. The Belgium *Training Fund for Research in Industry and Agriculture* (FRIA – F.R.S./FNRS) funded J.M.R. The forest companies Pallisco (Pasquet Group, Douala, Cameroon), ALPICAM (ALPI Group, Douala, Cameroon), ALPI Pietro and Sons (Kika, Cameroon), SFID-Mbang (Rougier Group, Douala, Cameroon), Mokabi SA (Rougier Group, Impfondo, Rep. of Congo), BPL-Lopola (Lopola, Rep. of Congo), CIB (OLAM Group, Brazzaville, Rep. of Congo), IFO (Danzer Group, Ouesso, Rep. of Congo), IFB (CAR), SOFOKAD (CAR), Thanry - TCA (VicWood, CAR), SCAF (Fadoul Group, CAR), and the NGO Nature+ (Belgium) provided forest inventories and logistical support during fieldwork. Maaike De Ridder (RMCA, Belgium) provided raw dendrochronological data for age modeling. We thank Achille Biwolé, Anaïs-Pasiphaé Gorel, Émile Fonty, Catherine Charles, Jérôme Perin and Gauthier Ligot (ULg/GxABT, Belgium) for their expertise in dendrometry and data analyses. We acknowledge Laura Maréchal (ULB, Belgium) for her help with the sample selection for dating and Ray Kidd, Christopher Bronk Ramsey (Univ. Oxford) and other OxCal group members for useful comments on modeling the radiocarbon dates. We also thank Valéry Gond (CIRAD) for his permission to use the MODIS map. Finally, Els Cornelissen and Nadine Devleeschouwer (RMCA, Belgium) helped in accessing the paleoenvironmental and archaeological literature, and Alexandre Livingstone Smith (RMCA, Belgium) added critical comments on the archaeological phases.

## Additional information

### Funding

| Funder | Author |
| --- | --- |
| Fonds pour la Formation à la Recherche dans l'Industrie et dans l'Agriculture | Julie Morin-Rivat |
| King Léopold III Fund for Nature Exploration and Conservation | Julie Morin-Rivat |
| Fonds De La Recherche Scientifique - FNRS | Julie Morin-Rivat<br>Adeline Fayolle<br>Hans Beeckman<br>Jean-Louis Doucet |
| EraNet-Biodiversa | Adeline Fayolle<br>Charly Favier<br>Laurent Bremond<br>Sylvie Gourlet-Fleury<br>Jean-Louis Doucet |
| Agence Nationale de la Recherche | Adeline Fayolle<br>Charly Favier<br>Laurent Bremond<br>Sylvie Gourlet-Fleury<br>Nicolas Bayol<br>Hans Beeckman |
| NERC Environmental Bioinformatics Centre | Adeline Fayolle<br>Charly Favier<br>Laurent Bremond<br>Sylvie Gourlet-Fleury<br>Nicolas Bayol<br>Jean-Louis Doucet |

The funders had no role in study design, data collection and interpretation, or the decision to submit the work for publication.

## Author contributions

JM-R, AF, J-LD, Conception and design, Acquisition of data, Analysis and interpretation of data, Drafting or revising the article, Contributed unpublished essential data or reagents; CF, LB, HB, Acquisition of data, Analysis and interpretation of data, Drafting or revising the article; SG-F, Acquisition of data, Analysis and interpretation of data, Drafting or revising the article, Contributed unpublished essential data or reagents; NB, Acquisition of data, Drafting or revising the article, Contributed unpublished essential data or reagents; PL, Analysis and interpretation of data, Drafting or revising the article

## Author ORCIDs

Julie Morin-Rivat, http://orcid.org/0000-0003-1823-6532

# Additional files

## Supplementary files

• Supplementary file 1. Trait information and characteristics of the diameter distribution for the 176 study genera across the SRI. Botanical family was extracted from the African Plant Database of the Conservatoire et Jardin botaniques de la Ville de Genève and South African National Biodiversity Institute, Pretoria (http://www.ville-ge.ch/musinfo/bd/cjb/africa/recherche.php). Trait information, including regeneration guild *sensu* Hawthorne (1995) (P = pioneers; NPLD = non-pioneer light-demanders; SB = shade-bearers), leaf phenology (deci = deciduous; ever = evergreen) and wood basic density, was extracted from Fayolle et al. (2014b). Diameter distribution was studied across the SRI using an ordination followed by a clustering (*Figure 2*). Total number of stems, mean diameter (in cm) and basal area (in m$^2$) are given. The four genera that are monospecific in the SRI and were used for age estimations are highlighted.

• Supplementary file 2. Age estimations of the trees at the mode of the diameter distribution for the four genera that are monospecific in the SRI. The mode of the diameter distribution across the SRI, information on growth data, including number of trees (n), the diameter (dbh) range and the mean and standard error of the annual diameter increment (SE), and age estimations of the mode based on the mean annual diameter increment (MAI$_d$) calculated for 982 monitored trees in the SRI are provided for the four study species (see *Figure 2—figure supplement 2*). For all study species, eight growth models (i.e., Canham, Gompertz, Verhulst, Power, Power mult, Lognormal, Linear and Mean) were fitted to the data, and age estimations were obtained with numerical solutions to ordinary differential equations (ODE) (see *Figure 2—figure supplement 2*). The most reliable age estimations according to age data from published tree-ring studies are highlighted.

• Supplementary file 3. Age data for the four study species based on published tree-ring data. Mean ages and corresponding estimated dates are shown in *Figure 2* (Main Text). C = Cameroon; RC = Republic of the Congo; CAR = Central African Republic; n = number of stem discs.

• Supplementary file 4. Relative performance of commonly used growth models for the four genera that are monospecific in the SRI. Growth model functions used to analyze the variation in tree growth (MAI$_d$, in cm.yr$^{-1}$) with tree size (DBH, in cm) are detailed below. For the biological interpretation of parameters, Max is the maximum growth or growth optimum (in cm.yr$^{-1}$), D$_{opt}$ is the diameter at growth optimum (in cm), and D$_{max}$ is the maximum diameter (in cm). Linear and mean models were additionally fitted to the data. For each model fitted to the growth and diameter data of each species, the Bayesian Information Criterion (BIC) is provided. The models with the best performance for each species are highlighted.

• Supplementary file 5. Data documenting paleoenvironmental changes (*Figures 1* and *3*, Main Text) during the last 1000 years in the SRI. C = Cameroon; RC = Republic of the Congo; CAR = Central

African Republic; W Africa = West Africa. Numbers refer to the map (*Figure 1* Main Text). References are indicated.

• Supplementary file 6. Synthesis of the 63 AMS radiocarbon and the two OSL dates documenting human activities (*Figure 3* Main Text) during the last 1000 years in the SRI. C = Cameroon; RC = Republic of the Congo; CAR = Central African Republic; AA = AMS Laboratory, University of Arizona (USA); Beta = Beta Analytic (USA); Erl = Erlangen AMS Facility (Germany); Gif = Gif-sur-Yvette (France); KI = Kiel (Germany); KIA = Kiel AMS (Germany); Ly = University of Lyon (France); OBDY = ORSTOM Bondy (France); Poz = Poznań Radiocarbon Laboratory (Poland). Numbers refer to the map (*Figure 1* Main Text). Dated material, identified species, lab codes, raw and calibrated dates Before Present (BP) and Anno Domini (AD), source references, and pottery (i.e. potsherds), smelting (i.e. iron slags and identified *tuyères*), salt (Richard Oslisly pers. obs.) and charred oil palm endocarps evidence are indicated. We assumed that charred oil palm endocarps found in combination with potsherds were consumed during the occupation of the sites (Morin-Rivat et al., 2016).

• Supplementary file 7. SQL codes for the Bayesian analysis of the radiocarbon dates.

• Supplementary file 8. Chronology of the historical events from the beginning of the 15th century to the present occurring or influencing human populations in the SRI. Precise dates or time spans are related to local or more general events. References are indicated.

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
