## [Decision Letter]

Thank you for submitting your article "Present-day central African forest is a legacy of the 19th century human history" for consideration by *eLife*. Your article has been reviewed by two peer reviewers (Charlotte Miller, Reviewer #1, and Bernard Clist, Reviewer #2), and the evaluation has been overseen by a Reviewing Editor and Christian Hardtke as the Senior Editor.

The reviewers were both in strong support of the work, requesting relatively minor revisions to the manuscript. Their comments are below.

Reviewer 1 said: "This paper is well written, clear, and concise and was a pleasure to read. The conclusions and interpretations are relevant, interesting and justifiable from the data presented. I was particularly impressed with the figures and the attention to detail; all figures are clear and relevant to the presentation and the discussion of the results. I fully recommend this paper to be published in *eLife*, and in my opinion, minimal corrections are needed."

Revisions:

Reviewer #1:

Abstract – Please state what methods you used for the study in the Abstract, instead of just 'an interdisciplinary approach'.

Abstract – 'regeneration shortage of these populations'- sounds a little strange. Do you mean lack of regeneration?

Introduction – ‘heckled history of disturbances’ – heckled is not the correct word to use here.

Introduction, first paragraph – years BP (or yr BP).

Introduction, first paragraph – 'Specifically, a dry event around 2500 ya caused forest fragmentation, and the fragmented forest included patches of savanna'. Reference.

Introduction, first paragraph – years BP.

Introduction, first paragraph – Correct reference to Brnčić et al., 2009 throughout manuscript.

Introduction, first paragraph – is this because more humans were present in the region?

Introduction, first paragraph – check order of references for journal (maybe they need to be in date order).

Introduction, second paragraph – please change to, 'Until the recent studies of X and X…'

Figure 1. What do the colours on the vegetation map represent?

Subsection “The regional history of human activities”, first paragraph – change from y to years or yrs.

Subsection “The regional history of human activities”, end of first paragraph – add reference.

Subsection “The regional history of human activities”, second paragraph – plural of pollen is pollen, not pollens.

Subsection “The regional history of human activities”, second paragraph – related to fires and burning? These are the same things? Are these fires for definite anthropogenic? Or maybe natural?

Subsection “The regional history of human activities”, third paragraph – in the frequency.

Supplementary

Subsection “Study area” – seasonally wet? Is it wet in the winter (DJF) and dry in the summer (JJA)?

Subsection “Study area” – does domain need capitalising?

Subsection “Forest inventory data” – sp of concessions.

Figure 2—figure supplement 1 – can't read the yellow text label.

Reviewer #2:

A discussion of the paper can be organized around two topics: what happened some 2500 years ago and the occupation by man of the SRI and other areas of Central Africa through time, and the archaeological signature inside the SRI since 1000 years ago.

1) The severe dry climatic episode the paper discusses in the first two paragraphs of the Introduction, stopped around 2500 bp as evidenced from the Mopo Bai site where at circa 2580 bp Poaceae pollens are at 36% of the total while circa 2400 bp they drop down to 13% which is evidence for a retreat of savanna areas in that sector after 2500 bp. The reforestation was ongoing then [Bostoen et al. 2015 pages 357-358: – Bostoen (K.), Clist (B.), Doumenge (C.), Grollemund (R.), Hombert (J.-M.), Koni Muluwa (J.) and Maley (J.), 2015, Middle to Late Holocene Paleoclimatic Change and the Early Bantu Expansion in the Rain Forests of Western Central Africa, Current Anthropology, 56 (3), pp.354-384]. J. Maley has previously argued for the existence of two different dry climatic episodes, one around 4000 bp affecting the periphery of the Equatorial forest, the other at 2500 bp affecting the core of it. The data can be used another way, with a start at 4000 bp of the regional dessication ending around 2500 bp with then the largest savanna expanses in position; it would have been a single palaeoclimatic crisis, soon after a slow process of forest regeneration with pioneer trees would have started, i.e. already at 2500 bp.

Subsection “The regional history of human activities”, first paragraph – it is suggested there was a 'massive depopulation' between AD 950-1300 with almost no human activity there. This should be better balanced as only some archaeologists lend an ear to such an interpretation while others prefer to speak of a lack of fieldwork to explain the apparent lack of human presence. An ongoing work to be published in 2017 tends to demonstrate the latter to be the better case.

2) The archaeological signature inside the SRI since 1000 years ago.

The caption of Figure 1 speaks of 52 archaeological sites. [Supplementary-material SD6-data] lists 65 dates coming from 38 sites only. Explain the discrepancy.

A note explaining the relevance of the 4 columns of the table would be welcome. Pottery is plain enough as is smelting though one should specify if this is iron slag only, or tewels, or iron slag and tewels, or furnaces, etc. The difference is important. Salt sites: how were they identified, what is their material identification, what is their relationship with settlements? Oil palm, must mean carbonized endocarps I believe. Can we be more specific? An endocarp can be carbonized in a natural forest fire as following human consumption.

Making good use of the [Supplementary-material SD6-data] and considering the settlement (pottery) and the industrial evidence (iron smelting) one finds a somewhat slightly different picture than the one presented.

The sites having yielded pottery are only 13 not 15 (2 dates each time at sites n°18 and 23).

Using the calibrated dates of the 15 dates associated to pottery (2 OSL and 13 14C) one finds 3 dates between AD 800 and 1100, none between AD 1100 and 1300, 8 between AD 1300 and 1600, 2 at AD 1700-1800, none after that. Looking at Figure 1, the map, we find a minimum of 50 km between these *pottery* sites, though sites 15 and 16, then sites 27 and 28 are situated closer together. This is not a good geographical and chronological sampling to discuss ancient man use through the large area of the SRI for the last 1000 years.

Using now the *smelting* sites as referenced in the paper, one finds one at AD 1000-1100, none between AD 1100-1300, 6 at AD 1300-1400, none at AD 1400-1500, 2 at AD 1500-1700, and an extraordinary 17 at AD 1700-1900. But most of them come from work done by Lupo et al. and Moga on only 5 sites, and Lupo et al. processed 17 14C dates from a single site (!), Bagbaya (site n°34) to the extreme north-east of the SRI.

This severely stresses again it is dangerous to use dates and especially radiocarbon dates in a global way without first getting into details of their respective contexts and making an internal criticism. Work done by the present reviewer on a series of 51 new 14C dates led before using them in a synthesis to discard 5 of them due to bad context.

The observation made of 17 14C dates associated to smelting during the short period of AD 1700-1900 should make one think of the associated deforestation sometimes quite extensive due to human settlements and evidenced in southern Congo (see work by B. Pinçon, iron smelting vs. volume of charcoal needed). This has an immediate relationship with the main topic of this paper.

The decrease of human impact on the forests due to a village redistribution during colonial times along the roads newly created has varied in time according to where you are situated: on the border between Cameroon and Gabon it happened in the early 20th century (see end of 19th century German maps of Fang villages in the Equatorial forest compared to French maps showing villages moved to the roads; R. Pourtier 1989 Le Gabon: espace, histoire et société, L'Harmattan, Paris). In the former Bas-Congo province of the Democratic Republic of Congo this happened in the early 1950s.

---

## [Author Response]

[…] Reviewer #1:

Abstract – Please state what methods you used for the study in the Abstract, instead of just 'an interdisciplinary approach'.

It has been detailed.

Abstract – 'regeneration shortage of these populations'- sounds a little strange. Do you mean lack of regeneration?

It has been corrected.

Introduction – ‘heckled history of disturbances’ – heckled is not the correct word to use here.

It has been corrected in “unequal”.

Introduction, first paragraph – years BP (or yr BP).

It has been added.

Introduction, first paragraph – 'Specifically, a dry event around 2500 ya caused forest fragmentation, and the fragmented forest included patches of savanna'. Reference.

The reference has been added: (Maley, 2002).

*Introduction, first paragraph – years BP.*

It has been corrected throughout the text.

Introduction, first paragraph – Correct reference to Brnčić et al., 2009 throughout manuscript.

It has been corrected throughout the text.

Introduction, first paragraph – is this because more humans were present in the region?

It has been precised.

Introduction, first paragraph – check order of references for journal (maybe they need to be in date order).

It has been already checked before submission.

Introduction, second paragraph – please change to, 'Until the recent studies of X and X…'

It has been changed.

Figure 1. What do the colours on the vegetation map represent?

The colors on the reference-map have been added in the caption by groups of colors.

Subsection “The regional history of human activities”, first paragraph – change from y to years or yrs.

It has been changed throughout the text.

Subsection “The regional history of human activities”, end of first paragraph – add reference.

References have been added.

Subsection “The regional history of human activities”, second paragraph – plural of pollen is pollen, not pollens.

It has been corrected throughout the text and tables.

Subsection “The regional history of human activities”, second paragraph – related to fires and burning? These are the same things? Are these fires for definite anthropogenic? Or maybe natural?

It has been precised: fires as natural and burning as anthropogenic.

Subsection “The regional history of human activities”, third paragraph – in the frequency.

It has been corrected.

Supplementary

Subsection “Study area” – seasonally wet? Is it wet in the winter (DJF) and dry in the summer (JJA)?

It has been precised.

Subsection “Study area” – does domain need capitalising?

It has been corrected.

Subsection “Forest inventory data” – sp of concessions.

It has been corrected.

Figure 2—figure supplement 1 – can't read the yellow text label.

It has been changed in green.

Reviewer #2:

A discussion of the paper can be organized around two topics: what happened some 2500 years ago and the occupation by man of the SRI and other areas of Central Africa through time, and the archaeological signature inside the SRI since 1000 years ago.

The main topic of the paper was the potential influence of anthropogenic disturbances on forests since 1000 years ago, and more particularly since the nineteenth century. That is the reason why we did not develop the 2500 BP event in this paper, otherwise well described in previous publications (e.g. Maley, 2002; Lézine et al., 2013).

*1) The severe dry climatic episode the paper discusses in the first two paragraphs of the Introduction, stopped around 2500 bp as evidenced from the Mopo Bai site where at circa 2580 bp Poaceae pollens are at 36% of the total while circa 2400 bp they drop down to 13% which is evidence for a retreat of savanna areas in that sector after 2500 bp. The reforestation was ongoing then [Bostoen et al. 2015 pages 357-358: – Bostoen (K.), Clist (B.), Doumenge (C.), Grollemund (R.), Hombert (J.-M.), Koni Muluwa (J.) and Maley (J.), 2015, Middle to Late Holocene Paleoclimatic Change and the Early Bantu Expansion in the Rain Forests of Western Central Africa, Current Anthropology, 56 (*3*), pp.354-384].*

This reference has been added.

J. Maley has previously argued for the existence of two different dry climatic episodes, one around 4000 bp affecting the periphery of the Equatorial forest, the other at 2500 bp affecting the core of it. The data can be used another way, with a start at 4000 bp of the regional dessication ending around 2500 bp with then the largest savanna expanses in position; it would have been a single palaeoclimatic crisis, soon after a slow process of forest regeneration with pioneer trees would have started, i.e. already at 2500 bp.

Indeed, this could be another way to interpret the data. Some colleagues and I are currently writing a project regarding savanna extension over time so as to verify this point.

Subsection “The regional history of human activities”, first paragraph – it is suggested there was a 'massive depopulation' between AD 950-1300 with almost no human activity there. This should be better balanced as only some archaeologists lend an ear to such an interpretation while others prefer to speak of a lack of fieldwork to explain the apparent lack of human presence. An ongoing work to be published in 2017 tends to demonstrate the latter to be the better case.

Thank you for sharing this information! Indeed, there is a crucial need for new archaeological and ^14^C data coming from central Africa. At the moment, and with regard of the data we had when writing this paper (completed with new field data collected in the SRI), we cannot say anything else than a lack of human activity during this time period, independently of the place investigated. I will be very happy to read this 2017 publication that will surely challenge our paper!

2) The archaeological signature inside the SRI since 1000 years ago.

The caption of Figure 1 speaks of 52 archaeological sites. [Supplementary-material SD6-data] lists 65 dates coming from 38 sites only. Explain the discrepancy.

It has been corrected throughout the manuscript.

A note explaining the relevance of the 4 columns of the table would be welcome. Pottery is plain enough as is smelting though one should specify if this is iron slag only, or tewels, or iron slag and tewels, or furnaces, etc. The difference is important.

It has been precised in the table caption.

Salt sites: how were they identified, what is their material identification, what is their relationship with settlements?

Richard Oslisly identified salt exploitation on the basis of current people habits and pottery shapes (to be published elsewhere).

Oil palm, must mean carbonized endocarps I believe.

I have added “charred” before “oil palm endocarps”.

Can we be more specific? An endocarp can be carbonized in a natural forest fire as following human consumption.

Charred oil palm endocarps were preferentially found in combination with potsherds, as analyzed in Morin-Rivat et al. 2016 (The Holocene). Obviously, endocarps could be charred naturally. The mention of the artifacts and ecofacts in [Supplementary-material SD6-data] are only indicative, as all publications we used did not systematically mention such findings.

Making good use of the [Supplementary-material SD6-data] and considering the settlement (pottery) and the industrial evidence (iron smelting) one finds a somewhat slightly different picture than the one presented.

Your comment suggests that settlements for living and craft places for smelting are not related to each other, or are far apart, depending of ore availability. [Supplementary-material SD6-data] gathers all evidence we have, independently from their function, either residential, or industrial, so as to provide a picture of the state of the art in the region. At the moment, this is rough, I confess. But fortunately, I have recently read about new discoveries of smelting sites (unpublished field report) nearby our own discoveries of residential sites, which could surely complete our understanding of site functions and site interactions in the SRI.

The sites having yielded pottery are only 13 not 15 (2 dates each time at sites n°18 and 23).

The paper does not mention such number about pottery. Maybe you refer to the sentence “15 dates from 13 sites in the Republic of the Congo” in the Supplementary material(?).

Using the calibrated dates of the 15 dates associated to pottery (2 OSL and 13 14C) one finds 3 dates between AD 800 and 1100, none between AD 1100 and 1300, 8 between AD 1300 and 1600, 2 at AD 1700-1800, none after that.

The Bayesian analysis used all dates but OSL dates, not only the dates associated with pottery. This modeling is a first step to picture out the intensity of human activities in the SRI.

Looking at Figure 1, the map, we find a minimum of 50 km between these pottery sites, though sites 15 and 16, then sites 27 and 28 are situated closer together. This is not a good geographical and chronological sampling to discuss ancient man use through the large area of the SRI for the last 1000 years.

This is a state of the art, and of course there is a huge gap i.e. lack of archaeological investigation in the SRI, particularly from Souanké to Berberati. The field report I mentioned above fill in a substantial part of this gap (approximately from the Dja River to Yokadouma). I wish these new results would be published soon.

Using now the smelting sites as referenced in the paper, one finds one at AD 1000-1100, none between AD 1100-1300, 6 at AD 1300-1400, none at AD 1400-1500, 2 at AD 1500-1700, and an extraordinary 17 at AD 1700-1900. But most of them come from work done by Lupo et al. and Moga on only 5 sites, and Lupo et al. processed 17 14C dates from a single site (!), Bagbaya (site n°34) to the extreme north-east of the SRI.

This severely stresses again it is dangerous to use dates and especially radiocarbon dates in a global way without first getting into details of their respective contexts and making an internal criticism.

This has been balanced in the conclusion.

Work done by the present reviewer on a series of 51 new 14C dates led before using them in a synthesis to discard 5 of them due to bad context.

We did not report bad contexts in the present study, although we challenged all of them. But, I agree with you about looking for such problems as prerequisites for radiocarbon analyses.

The observation made of 17 14C dates associated to smelting during the short period of AD 1700-1900 should make one think of the associated deforestation sometimes quite extensive due to human settlements and evidenced in southern Congo (see work by B. Pinçon, iron smelting vs. volume of charcoal needed). This has an immediate relationship with the main topic of this paper.

Of course, iron smelting had an environmental impact. This point has been precised in the part “Recent and generalized land abandonment”.

However, as mentioned by Lupo et al. (2015), there is still a debate about evaluating the volume of wood needed for metallurgy, compared to the volume of trees logged for shifting agriculture (by referring to Goucher 1981, and Fairland and Leach 1998 publications). In the present paper, we did not deny the potential important environmental impacts of both activities. They were simply grouped into the single category “human activities”. More work should be done regarding site functions and site spatial influence.

The decrease of human impact on the forests due to a village redistribution during colonial times along the roads newly created has varied in time according to where you are situated: on the border between Cameroon and Gabon it happened in the early 20th century (see end of 19th century German maps of Fang villages in the Equatorial forest compared to French maps showing villages moved to the roads; R. Pourtier 1989 Le Gabon: espace, histoire et société, L'Harmattan, Paris). In the former Bas-Congo province of the Democratic Republic of Congo this happened in the early 1950s.

Thank you for this reference. Indeed, I think that mortality (i.e. because of conflicts, hard work, and diseases) was the first cause of population decrease in the last decades of the nineteenth century, and in the first of the twentieth. As mentioned in our paper, during the following years (since the 1920s, as proposed in [Supplementary-material SD7-data]), the organization of the colonial administration and trade, then the rural exodus where very important causes of this forest depopulation.

I had access to the rare French and German maps dated to the end of the nineteenth century, and the situation in the SRI was less documented than that for Gabon. I wish I could find more maps in the national archives, including the Belgian colonial ones for western DRC (not accessible yet because of an ongoing transfer to the Royal Library).